# Critic Learning-Based Safe Optimal Control for Nonlinear Systems with Asymmetric Input Constraints and Unmatched Disturbances

**DOI:** 10.3390/e25071101

**Published:** 2023-07-24

**Authors:** Chunbin Qin, Kaijun Jiang, Jishi Zhang, Tianzeng Zhu

**Affiliations:** 1School of Artificial Intelligence, Henan University, Zhengzhou 450000, China; qcb@henu.edu.cn (C.Q.); 1824100055@vip.henu.edu.cn (K.J.); kz520@henu.edu.cn (T.Z.); 2School of Software, Henan University, Kaifeng 475000, China

**Keywords:** critic neural network, asymmetric input constraints, unmatched disturbances, safety, adaptive dynamic programming, nonlinear systems

## Abstract

In this paper, the safe optimal control method for continuous-time (CT) nonlinear safety-critical systems with asymmetric input constraints and unmatched disturbances based on the adaptive dynamic programming (ADP) is investigated. Initially, a new non-quadratic form function is implemented to effectively handle the asymmetric input constraints. Subsequently, the safe optimal control problem is transformed into a two-player zero-sum game (ZSG) problem to suppress the influence of unmatched disturbances, and a new Hamilton–Jacobi–Isaacs (HJI) equation is introduced by integrating the control barrier function (CBF) with the cost function to penalize unsafe behavior. Moreover, a damping factor is embedded in the CBF to balance safety and optimality. To obtain a safe optimal controller, only one critic neural network (CNN) is utilized to tackle the complex HJI equation, leading to a decreased computational load in contrast to the utilization of the conventional actor–critic network. Then, the system state and the parameters of the CNN are uniformly ultimately bounded (UUB) through the application of the Lyapunov stability method. Lastly, two examples are presented to confirm the efficacy of the presented approach.

## 1. Introduction

Safety-critical systems are those that, in case of accidents or failures, can result in significant consequences, including but not limited to injuries, loss of life, environmental harm, or financial losses. The emergence of safety-critical systems like unmanned aerial vehicles (UAVs) [1,2,3] and robots [4] has led to an increased focus on safety control design within the field of control systems [5,6]. Safety control designs entail control strategies that satisfy safety specifications imposed by environmental limitations or physical limitations of the system. Ignoring the detrimental impact of safety entails substantial risks to both the safety of belongings and personal security. To address the challenges of the safe controller design, researchers have provided some effective approaches [7,8,9,10,11]. The problem of safety in the presence of unmodeled dynamics or disturbances in drones has recently been addressed by designing the robust controller based on the nonlinear estimator in [9]. In ref. [10], the use of neural networks integrated with the Lyapunov theory was preliminarily treated with application in the automotive sector for critical situations, and this aspect was further addressed in an even more organic way. In ref. [11], the quadratic programming-based method was applied to develop a safe controller. Despite the fact that this method can guarantee safety at a local level for every time step, selecting a step size that is too small leads to redundant computations. In contrast, a step size that is too large causes unsafe behavior, making it challenging to ensure the safety of the system. Hence, it is crucial to identify an appropriate control design method for CT safety-critical systems that can guarantee the safety of the systems.

Recently, the CBF technique has emerged as an effective approach for ensuring the security of safety-critical systems [12,13,14]. The underlying principle of the CBF is to insure the forward invariance of the safe set. In ref. [15], the safe-based reinforcement learning approach was demonstrated, where the CBF was merged into the cost function to assure both the safety and optimality of the system. Typically, the CBF component is contained within the primitive cost function to penalize behavior that violates safety constraints. Reference [16] incorporated damping factors into the CBF and intervened selectively only in the event of safety constraint violations, aiming to reduce disruptions to the optimal controller. Reference [17] introduced the utilization of the CBF and summarized the verification approach for safety-critical control systems. Nevertheless, the methods mentioned above do not take into account the presence of external disturbances, which served as a motivation for the research conducted in this paper.

As disturbances are present in almost all industrial systems and hurt control performance, it is necessary to consider external disturbances in actual projects. Recently, several methods have been proposed to address disturbances [18,19,20,21,22,23,24]. For instance, references [18,19] used H∞ control to reduce external disturbances in nonlinear systems. Reference [23] combined ADP with sliding mode control for addressing optimal control problems of CT nonlinear systems considering uncertain disturbances. Reference [24] cast external disturbances as the ZSG problem, with the control strategy aiming at minimizing the cost function and the disturbance strategy striving towards maximizing it. It is well known that for the HJI equation of the ZSG, it is difficult to find its analytical solution. Fortunately, with the evolution of optimal control [25,26,27], the ADP approach [28] was employed to approximately tackle the ZSG problem. For example, in reference [29], a new database-based adaptive critic algorithm was presented to study the infinite-scale robust control for nonlinear systems. However, the aforementioned methods fail to consider the capability limitations of the system due to asymmetric input constraints.

Although symmetric input constraints have been widely investigated in prior research and tackled with various techniques, such as the control problem that concerns the uncertain impulse system that has input constraints, which was handled in [30], and the utilization of integral reinforcement learning with the actor–critic network to address the tracking control problem under input constraints in [31,32], there has been relatively little research on the treatment of asymmetric input constraints that frequently occur in practical systems. Several optimal control methods exist for addressing CT systems with nonlinear dynamics and input constraints that are asymmetric. Among them is one that used the cost function with adjustable upper and lower limits of integration [33,34,35]. Another proposed the switching function [36] to tackle the problem, but it is only applied to linear systems. However, none of these results considered the incorporation of CBF into the CT nonlinear safety-critical systems to study the safe and optimal control problem under asymmetric input constraints with external disturbances.

This study explores the safe and optimal control issue for safety-critical nonlinear systems to reject unmatched external disturbances under the condition of asymmetric input constraints. Unlike other works, a new non-quadratic form function for handling asymmetric input constraints is proposed in this paper. To tackle the challenge posed by unmatched disturbances, a two-player ZSG is put forward to formulate the optimization problem. The ZSG is then addressed by finding the Nash equilibrium point, which is obtained by addressing the HJI equation. However, since solving the HJI equation is challenging, an ADP technique similar to that used in references [37,38,39] is exploited to estimate the solution of the HJI equation. In addition, one single CNN is used instead of a dual actor–critic neural network to diminish the computational complexity in approximating the control policy. Consequently, the optimal control policy is obtained by considering the worst disturbance.

The contributions are outlined mainly as follows:Asymmetric input constraints are considered in the control problem of the CT nonlinear safety-critical systems. In addition, this paper proposes a new non-quadratic form function to address the issue of asymmetric input constraints. It is important to note that when applying this approach, the optimal control policy no longer remains at 0, even when the system state reaches the equilibrium point of x=0 (see $u^{*}\left(x\right)$ in later Equation (Equation 15)).This paper adopts the CBF to construct safety constraints and proposes designing a damping coefficient within the CBF to balance the safety and optimality of safety-critical systems based on varying safety requirements in different applications.The safe optimal control problem is turned into the ZSG problem to address unmatched disturbances; then, the optimal control law is gained by tackling the HJI equation using one CNN. Moreover, the use of only one CNN to approximate the HJI equation is an effective way to reduce the computational burden compared to the actor–critic network and the system state, and CNN parameters are demonstrated to be UUB.

The following structure is adopted for this article. Section 2 provides the initial formulation of the problem. Section 3 presents a safe optimal control design for the two-player ZSG problem. Then, in Section 4, an adaptive CNN method for addressing the HJI equation using an online method is proposed, and its stability is verified. Section 5 introduces two examples to demonstrate that the presented approach is effective. Lastly, Section 6 gives conclusions.

## 2. Problem Statement

Consider the CT nonlinear safety-critical system as
(1)$$\dot{x}=F\left(x\right)+G\left(x\right)u+P\left(x\right)v,$$
where $x={\left[x_{1},x_{2},\ldots ,x_{n}\right]}^{T}\in C_{a}\subseteq R^{n}$ indicates the system state vector with *n*-dimensional parameters, $F\left(x\right)\in R^{n}$ represents the internal dynamics, $G\left(x\right)\in R^{n\times m}$ and $P\left(x\right)\in R^{n\times q}$ indicate control and disturbance coefficient matrices, respectively. Additionally, $u\in R^{m}$ denotes an input variable with *m*-dimensional parameters denoted by ${∁}_{u}=\left\{u|u_{max}\geq u\geq u_{min}\right\}$, where $u_{max}$ and $u_{min}$ stand for the upper and lower bounds, respectively. And $v\in R^{q}$ is the unmatched disturbances. The paper assumes F(·), G(·), P(·) are Lipschitz continuous and satisfy F(0)=0, and the safety-critical System (Equation 1) is stabilizable and controllable. Moreover, we assume there exist two constants $G_{M}>0$ and $P_{M}>0$. Both G(x) and P(x) have upper bounded values, i.e., $G_{M}\geq ∥G(x)∥,P_{M}\geq ∥P(x)∥$, for any $x\in R^{n}$.

In addition, it is essential to emphasize that $C_{a}$ represents a safe set for (Equation 1). $C_{a}$ is derived from operational restrictions, such as the allowable states of the robot arm, which is mathematically determined by
(2)$$\left\{\begin{array}{c} \begin{array}{cc} C_{a} & =\left\{x\in R^{n}|z\left(x\right)\geq 0\right\},\\ int(C_{a}) & =\left\{x\in R^{n}|z\left(x\right)>0\right\},\\ \partial C_{a} & =\left\{x\in R^{n}|z\left(x\right)=0\right\}, \end{array} \end{array}\right.$$
where z(x) represents continuous concerning *x*. The set $int(C_{a})$ denotes the interior of $C_{a}$, while $\partial C_{a}$ represents the boundary of $C_{a}$.

Subsequently, the representation of the infinite horizon cost function from t=0 for System (Equation 1) is given by
(3)$$V\left(x\right)=\int_{0}^{\infty }x{\left(t\right)}^{T}Qx\left(t\right)+U\left(u\right)-\Upsilon^{2}{∥v∥}^{2}dt,$$
where Q represents a function with positive definite properties, ${∥v∥}^{2}$ = $v^{T}v$, Υ>0 represents a constant weight coefficient, U(u) is a non-quadratic form function employed for handling the asymmetric input constraints determined by
(4)$$\begin{array}{cc} U(u) & =2\int_{ℑ}^{u}\Psi tanh^{-1}\left(\frac{t-ℑ}{\Psi }\right)dt\\ \; & =2\Psi \left(u-ℑ\right)tanh^{-1}\left(\frac{u-ℑ}{\Psi }\right)+\Psi^{2}\ln_{}(1-\frac{{\left(u-ℑ\right)}^{2}}{\Psi }), \end{array}$$
with Ψ and *ℑ* defined as
(5)$$\Psi =\frac{1}{2}\left(u_{max}-u_{min}\right),ℑ=\frac{1}{2}\left(u_{max}+u_{min}\right),$$
where $\left|u_{max}\right|\neq \left|u_{min}\right|$ and $tanh\left(z\right)=\left(e^{z}-e^{-z}\right)/\left(e^{z}+e^{-z}\right)$ with z∈R.

**Remark 1.** 
*Even though tanh(z) is symmetric, U(u) in (Equation 4) generates asymmetric constraints in the control signal $u^{*}\left(x\right)$ (see $u^{*}\left(x\right)$ in later (Equation 15)). This is due to the fact that ℑ is not equal to 0 in (Equation 4). This feature is different from studying the symmetric input constraints.*


Additionally, the ultimate objective of this paper is to devise the safe and optimal control input policy for (Equation 1), which involves the utilization of the CBF concept. In the upcoming section, this paper presents the concept of the CBF and proposes an ADP-based approach to design the safe and optimal controller.

## 3. Safe Optimal Control Design

This section presents a detailed explanation of the concept of the CBF. Then, the safe and optimal control problem is converted to the two-player ZSG to overcome the unmatched disturbances, and the CBF is integrated with the cost function without an intermediary to punish unsafe behavior.

### 3.1. Control Barrier Function

The utilization of the CBF provides a solution to address the safety constraint problem in safety-critical systems. The CBF is a function that is non-negative within the set $C_{a}$ and exhibits divergence to infinity at the edge of $C_{a}$. As the state *x* is about to reach the boundary of $C_{a}$, the condition of negative derivative can bring the system state *x* back within $C_{a}$, ensuring that the system state is always confined within $C_{a}$. To better illustrate the properties of the CBF, the following assumption is given.

**Assumption 1.** 
*The CBF candidate $B_{r}\left(x\right)$ meets the subsequent three characteristics [40,41]:*
*(1)* 
*$B_{r}\left(x\right)\geq 0,\forall x\in int\left(C_{a}\right)$,*
*(2)* 
*$B_{r}\left(x\right)\to \infty ,\forall x\in \partial C_{a}$,*
*(3)* 
*$B_{r}\left(x\right)$ is monotonically decreasing $\forall x\in C_{a}$.*



Moreover, for all $x\in C_{a}$, the CBF $B_{r}\left(x\right)$ has the following properties: (6)$$\begin{array}{c} \frac{1}{\gamma_{1}\left(z\left(x\right)\right)}\leq B_{r}\left(x\right)\leq \frac{1}{\gamma_{2}\left(z\left(x\right)\right)},\dot{B_{r}}\left(x\right)\leq \gamma_{3}\left(z\left(x\right)\right), \end{array}$$
where $\gamma_{1}\left(\cdot \right),\gamma_{2}\left(\cdot \right)$, and $\gamma_{3}\left(\cdot \right)$ are class K functions.

Under the premise that Assumption 1 and Equation (Equation 6) both hold, a suitable choice for $B_{r}\left(x\right)$ is ρy(x)/z(x), where y(x) represents a special scheduling function determined by the user to allow for flexibility in selecting $B_{r}\left(x\right)$. Specifically, y(x) ensures that the CBF operates only when the system is close to the unsafe set. ρ>0 is the damping factor used to balance safety and optimality.

**Remark 2.** 
*In contrast to the previous CBF *[16]*, the ρ chosen here shows a positive correlation with the value of $B_{r}\left(x\right)$. The larger the value of ρ, the faster the system state moves away from the unsafe set, and the smaller the value of ρ, the slower the state x moves away from the unsafe set. A smaller value of ρ emphasizes optimality and a larger value of ρ enforces safety.*


### 3.2. Safe and Optimal Control Approach

By augmenting the selected CBF $B_{r}\left(x\right)$ to the cost function (Equation 3), a new refined cost function is obtained, that is,
(7)$$\begin{array}{c} V\left(x\right)=\int_{0}^{\infty }x{\left(t\right)}^{T}Qx\left(t\right)+U\left(u\right)-\Upsilon^{2}{∥v∥}^{2}+B_{r}\left(x\right)dt. \end{array}$$

**Remark 3.** 
*To ensure the safety of the system, it is assumed that the original system state x is confined within the set $C_{a}$. This is because the rapid increase in $B_{r}\left(x\right)$ as the state x nears the boundary of $C_{a}$ is the reason behind the penalization of state convergence behavior when the initial state is beyond $C_{a}$. This prevents the system state from converging.*


The conventional control problems can be transformed into two-player ZSG problems. The Nash equilibrium point, i.e., the saddle point ($u^{*}$,$v^{*}$) can be obtained by addressing the special HJI equation. Then, the optimal cost function is defined by
(8)$$\begin{array}{c} V^{*}\left(x\right)=\underset{u}{min}\;\underset{v}{max}\int_{0}^{\infty }x{\left(t\right)}^{T}Qx\left(t\right)+U\left(u\right)-Y^{2}{∥v∥}^{2}+B_{r}\left(x\right)dt. \end{array}$$

The purpose of the two-player ZSG problem is to identify a saddle point so that the following inequality can hold: (9)$$\begin{array}{c} V^{*}\left(x,u^{*},v\right)\leq V^{*}\left(x,u^{*},v^{*}\right)\leq V^{*}\left(x,u,v^{*}\right). \end{array}$$

Therefore, for the two-player ZSG problem, $u^{*}$ is the optimal control input policy minimizing the cost function, and $v^{*}$ represents the worst disturbance input policy maximizing the cost function.

**Definition 1.** 
*Input policy u is considered admissible in relation to (Equation 7) on $℧\in R^{n}$, denoted by u∈ℵ(℧), u stabilizes (Equation 1) on ℧ if u is continuous on ℧, and (Equation 7) is limited for any x∈℧.*


For the admissible input policy u∈ℵ(℧), if Equation (Equation 7) is continuously differentiable, computing the gradient of V(x) with respect to *t* on both sides of Equation (Equation 7) yields the nonlinear Lyapunov equation as
(10)$$\begin{array}{c} 0=\nabla V{\left(x\right)}^{T}\left(F\left(x\right)+G\left(x\right)u+P\left(x\right)v\right)+x{\left(t\right)}^{T}Qx\left(t\right)+U\left(u\right)-Y^{2}{∥v∥}^{2}+B_{r}\left(x\right), \end{array}$$
where ∇V(x) is the gradient of V(x), V(0)=0.

Based on the optimal control approach, the HJI equation for the two-player ZSG problem possesses an exclusive solution if there exists a saddle point, that is, if the following conditions hold: (11)$$\begin{array}{c} 0=\underset{u}{min}\;\underset{v}{max}H\left(x,u,v,\nabla V^{*}\left(x\right)\right)=\underset{v}{max}\;\underset{u}{min}H\left(x,u,v,\nabla V^{*}\left(x\right)\right), \end{array}$$
where $H(x,u,v,\nabla V^{*}\left(x\right))$ refers to the Hamiltonian function of the safety-critical system (Equation 1), that is,
(12)$$\begin{array}{cc} H(x,u,v,\nabla V^{*}\left(x\right))=\nabla & V^{*}{\left(x\right)}^{T}\left(F\left(x\right)+G\left(x\right)u+P\left(x\right)v\right)\\ \; & +x{\left(t\right)}^{T}Qx\left(t\right)+U\left(u\right)-Y^{2}{∥v∥}^{2}+B_{r}\left(x\right). \end{array}$$

By using Equations (Equation 11) and (Equation 12), the saddle point can be found by addressing two equations as
(13)$$u^{*}\left(x\right)=arg\;\underset{u}{min}H\left(x,u,v,\nabla V^{*}\left(x\right)\right),$$
and
(14)$$v^{*}\left(x\right)=arg\;\underset{v}{max}H\left(x,u,v,\nabla V^{*}\left(x\right)\right).$$

Thus, the saddle point $\left(u^{*},v^{*}\right)$ can be gained as
(15)$$u^{*}\left(x\right)=-\Psi \;tanh\left(\frac{1}{2\Psi }G{\left(x\right)}^{T}\nabla V^{*}\left(x\right)\right)+\psi_{ℑ},$$
and
(16)$$v^{*}\left(x\right)=\frac{1}{2Y^{2}}P{\left(x\right)}^{T}\nabla V^{*}\left(x\right),$$
where $\psi_{ℑ}={\left[ℑ,ℑ,\ldots ,ℑ\right]}^{T}\in R^{m}$ with *ℑ* given by Equation (Equation 5).

**Remark 4.** 
*Given that ℑ≠0 from (Equation 5), it can be concluded that u(0)=ℑ≠0. Therefore, in order to establish the equilibrium point of (Equation 1) at x=0, the assumption of G(0)=0 is necessary.*


Substituting Equations (Equation 15) and (Equation 16) into Equation (Equation 11), the HJI equation can be redefined as
(17)$$\begin{array}{cc} 0= & \nabla V^{*}{\left(x\right)}^{T}\left(F\left(x\right)+P\left(x\right)v^{*}\right)+x{\left(t\right)}^{T}Qx\left(t\right)+U\left(-\Psi \;tanh\left(T\left(x\right)\right)+\psi_{ℑ}\right)\\ \; & -Y^{2}{∥v^{*}∥}^{2}-{\left(\Psi \nabla V^{*}\left(x\right)\right)}^{T}G\left(x\right)tanh\left(T\left(x\right)\right)+\nabla V^{*}{\left(x\right)}^{T}G\left(x\right)\psi_{ℑ}+B_{r}\left(x\right), \end{array}$$
where $T\left(x\right)=1/\left(2\Psi \right)G{\left(x\right)}^{T}\nabla V^{*}\left(x\right)$ and $V^{*}\left(0\right)=0$.

For the optimal safe control problem of the ZSG with unmatched external disturbances and asymmetric input constraints, it is necessary to obtain the value corresponding to the optimal cost Function (Equation 8) for achieving the optimal control input Policy (Equation 15) and the worst disturbance input Policy (Equation 16). Therefore, the solution of Equation (Equation 17) needs to be obtained. Nevertheless, since Equation (Equation 17) represents a nonlinear partial differential equation, it is challenging to find its analytical solution using conventional mathematical approaches. Hence, the solution of this equation is estimated by using the CNN in the next section.

## 4. Adaptive CNN Design

### 4.1. Solving the HJI Equation via the CNN

This section designs a CNN to estimate cost function $V^{*}\left(x\right)$ as
(18)$$V^{*}\left(x\right)=W_{c}^{T}\delta \left(x\right)+\xi \left(x\right),$$
where ξ(x) represents the estimation error about the CNN with ξ(0)=0, $W_{c}\in R^{r}$ represents the ideal weight vector of the CNN, $\delta \left(x\right)=[\delta_{1}\left(x\right);\delta_{2}\left(x\right);\ldots ;\delta_{r}\left(x\right)]$ represents activation function with $\delta_{j}\left(0\right)=0,j=1,2,\ldots ,r,r$ is the number of neurons in the CNN.

The gradient of the approximate optimal cost function is
(19)$$\nabla V^{*}\left(x\right)=\nabla \delta {\left(x\right)}^{T}W_{c}+\nabla \xi \left(x\right).$$

Substituting Equation (Equation 19) into Equation (Equation 15), $u^{*}\left(x\right)$ can be represented as
(20)$$u^{*}\left(x\right)=-\Psi \;tanh\left(\bar{A}\left(x\right)\right)+\xi_{u^{*}}\left(x\right)+\psi_{ℑ},$$
where
(21)$$\bar{A}\left(x\right)=\frac{1}{2\Psi }G{\left(x\right)}^{T}\nabla \delta {\left(x\right)}^{T}W_{c},$$
and
(22)$$\xi_{u^{*}}\left(x\right)=-\frac{1}{2}\left(I_{m}-\Phi \left(A\left(x\right)\right)\right)G{\left(x\right)}^{T}\nabla \xi \left(x\right),$$
with $\Phi \left(A\left(x\right)\right)=diag\left\{tanh^{2}\left(A_{l}\left(x\right)\right)\right\}\left(l=1,2,\ldots ,m\right)$ with $A_{l}\left(x\right)=[A_{1}\left(x\right);A_{2}\left(x\right);\ldots ;$$A_{m}\left(x\right)]\in R^{m}$ being selected between $\bar{A}\left(x\right)$ and T(x). Then, considering Equation (Equation 19), $v^{*}\left(x\right)$ in Equation (Equation 16) can be redefined as
(23)$$v^{*}\left(x\right)=\frac{1}{2Y^{2}}P{\left(x\right)}^{T}\nabla \delta {\left(x\right)}^{T}W_{c}+\xi_{v^{*}}\left(x\right),$$
where $\xi_{v^{*}}\left(x\right)=\frac{1}{2\Upsilon^{2}}P{\left(x\right)}^{T}\nabla \xi \left(x\right).$

Similarly, substituting Equation (Equation 19) into Equation (Equation 17), the HJI equation can be rewritten as
(24)$$\begin{array}{cc} 0= & W_{c}^{T}\nabla \delta \left(x\right)\left(F\left(x\right)+P\left(x\right)v^{*}\right)+x^{T}Qx+U\left(-\Psi tanh\left(\bar{A}\left(x\right)+K\left(x\right)+\psi_{ℑ}\right)\right)+B_{r}\left(x\right)\\ \; & -\Psi W_{c}^{T}\nabla \delta \left(x\right)G\left(x\right)tanh\left(\bar{A}\left(x\right)+K\left(x\right)\right)-\Psi \nabla \xi {\left(x\right)}^{T}G\left(x\right)tanh\left(\bar{A}\left(x\right)+K\left(x\right)\right)\\ \; & +\nabla \xi {\left(x\right)}^{T}\left(F\left(x\right)+P\left(x\right)v^{*}\right)-Y^{2}{∥v^{*}∥}^{2}+\left(W_{c}^{T}\nabla \delta \left(x\right)+\nabla \xi {\left(x\right)}^{T}\right)G\left(x\right)\psi_{ℑ}, \end{array}$$
where $K\left(x\right)=1/\left(2\Psi \right)G{\left(x\right)}^{T}\nabla \xi \left(x\right)$.

However, since the ideal CNN weight $W_{c}$ in Equation (Equation 18) is unknown, it can not be used in the control procedure. Hence, the CNN is used to estimate the cost function and its gradient as
(25)$$\hat{V}\left(x\right)={\hat{W}}_{c}^{T}\delta \left(x\right),$$
(26)$$\nabla \hat{V}\left(x\right)=\nabla \delta {\left(x\right)}^{T}{\hat{W}}_{c},$$
where ${\hat{W}}_{c}$ represents the estimation of $W_{c}$.

Therefore, the approximate optimal input and the approximate worst disturbance input become
(27)$${\hat{u}}^{*}\left(x\right)=-\Psi tanh\left(\frac{1}{2\Psi }G{\left(x\right)}^{T}\nabla \delta {\left(x\right)}^{T}{\hat{W}}_{c}\right)+\psi_{ℑ},$$
and
(28)$${\hat{v}}^{*}\left(x\right)=\frac{1}{2Y^{2}}G{\left(x\right)}^{T}\nabla \delta {\left(x\right)}^{T}{\hat{W}}_{c}.$$

Subsequently, the approximated Hamilton function can be formulated by
(29)$$\begin{array}{cc} \hat{H}\left(x,{\hat{W}}_{c},{\hat{v}}^{*}\right)= & {\hat{W}}_{c}^{T}ð+{\hat{W}}_{c}^{T}\nabla \delta \left(x\right)G\left(x\right)\psi_{ℑ}+U\left(-\Psi tanh\left(\Gamma \left(x\right)\right)+\psi_{ℑ}\right)\\ \; & +x^{T}Qx-Y^{2}{∥{\hat{W}}_{c}∥}^{2}-\Psi {\hat{W}}_{c}^{T}\nabla \delta \left(x\right)G\left(x\right)tanh\left(\Gamma \left(x\right)\right)+B_{r}\left(x\right), \end{array}$$
where
(30)$$ð=\nabla \delta \left(x\right)(F\left(x\right)+P\left(x\right){\hat{v}}^{*})$$
and
(31)$$\Gamma \left(x\right)=\frac{1}{2\Psi }G{\left(x\right)}^{T}\nabla \delta {\left(x\right)}^{T}{\hat{W}}_{c}.$$

The CNN weight estimation error is denoted by
(32)$${\tilde{W}}_{c}=W_{c}-{\hat{W}}_{c},$$
and the approximation error $\varrho_{c}$ of the Hamiltonian function is derived as
(33)$$\begin{array}{cc} \varrho_{c} & =\hat{H}\left(x,{\hat{W}}_{c},{\hat{v}}^{*}\right)-H\left(x,u^{*},v^{*},\nabla V^{*}\left(x\right)\right)\\ \; & =\hat{H}\left(x,{\hat{W}}_{c},{\hat{v}}^{*}\right). \end{array}$$

To achieve ${\hat{W}}_{c}\to W_{c}$, it is necessary to ensure that $\varrho_{c}\to 0$. Therefore, the chosen target function is denoted by $E=\frac{1}{2}\varrho_{c}^{T}\varrho_{c}\left(1/{\left(1+{ð}^{T}ð\right)}^{2}\right)$, where $O=1+{ð}^{T}ð$. Consequently, based on a normalized gradient descent algorithm, the weight vector ${\hat{W}}_{c}$ is defined by
(34)$${\dot{\hat{W}}}_{c}=-\frac{\alpha }{O^{2}}\frac{\partial E}{\partial {\hat{W}}_{c}}=-\frac{\alpha }{O^{2}}\varrho_{c},$$
with α>0 being the adjustable parameter and $\varrho_{c}$ defined as Equation (Equation 33).

Using Equations (Equation 32) and (Equation 34), the weight approximation error ${\dot{\tilde{W}}}_{c}$ can be expressed as
(35)$${\dot{\tilde{W}}}_{c}=\frac{\alpha \zeta }{O}\xi_{c}-\alpha \zeta \zeta^{T}{\tilde{W}}_{c},$$
where $\xi_{c}=-\nabla \xi {\left(x\right)}^{T}\left(F\left(x\right)+P\left(x\right){\hat{v}}^{*}\right)$ is the residual error and $\zeta =\frac{ð}{O}$.

### 4.2. Stability Analysis

The UUB of both the state *x* and the CNN parameters in the closed-loop system is demonstrated by utilizing the Lyapunov stability analysis principle in this subsection. First, two assumptions that were also used in [28,42] are required, as

**Assumption 2.** 
*The ideal optimal CNN weight vector $W_{c}$ is upper bounded, i.e., $∥W_{c}∥\leq b_{W_{c}}$, where $b_{W_{c}}>0$ is a constant. Moreover, for any x∈℧, this paper assumes that there are two known constants $b_{\nabla \delta }>0$, $b_{\delta }>0$ so that $∥\nabla \delta (x)∥\leq b_{\nabla \delta },\;∥\delta (x)∥\leq b_{\delta }$. Meanwhile, there exist $b_{\nabla \xi }>0$ and $b_{\xi }>0$ so that $∥\nabla \xi (x)∥\leq b_{\nabla \xi },∥\xi (x)∥\leq b_{\xi }$ for any x∈℧.*


**Assumption 3.** 
*We make $b_{\xi_{u^{*}}},b_{\xi_{v^{*}}},b_{\xi_{c}}$ be positive constants.*
*(1)* 
*$b_{\xi_{u^{*}}}\geq ∥\xi_{u^{*}}\left(x\right)∥$ for any x∈℧.*
*(2)* 
*$b_{\xi_{v^{*}}}\geq ∥\xi_{v^{*}}\left(x\right)∥$ for any x∈℧.*
*(3)* 
*$b_{\xi_{c}}\geq ∥\xi_{c}∥$ for any x∈℧.*



**Theorem 1.** 
*Assuming Assumptions 1–3 are met, we consider System (Equation 1) with the associated Control (Equation 27) and the update rule of CNN (Equation 34), ensuring all signals in the nonlinear system are UUB if the following condition holds:*

(36)
$$\alpha k_{min}\left(\zeta \zeta^{T}\right)-\left(1/Y^{2}\right){ℑ}_{\nabla \delta }^{2}P_{M}^{2}>0.$$



**Proof.** We let the Lyapunov candidate function as the following (note: for convenience, $V^{*}\left(x\right)$ and $\left(1/2\right){\tilde{W}}_{c}^{T}{\tilde{W}}_{c}$ are abbreviated as $L_{1}$ and $L_{2}$ below):
(37)$$L\left(t\right)=\underset{L_{1}}{\underbrace{V^{*}\left(x\right)}}+\underset{L_{2}}{\underbrace{\left(1/2\right){\tilde{W}}_{c}^{T}{\tilde{W}}_{c}}}.$$Taking the derivation of $L_{1}$ in Equation (Equation 37) and using System (Equation 1), the derivation of $L_{1}$ can be expressed as
(38)$$\begin{array}{cc} {\dot{L}}_{1} & =\frac{dV^{*}\left(x\right)}{dt}\\ \; & =\nabla V^{*}{\left(x\right)}^{T}\left(F\left(x\right)+G\left(x\right){\hat{u}}^{*}+P\left(x\right){\hat{v}}^{*}\right)\\ \; & =\nabla V^{*}{\left(x\right)}^{T}\left(F\left(x\right)+G\left(x\right)u^{*}+P\left(x\right)v^{*}\right)\\ \; & \;\;+\nabla V^{*}{\left(x\right)}^{T}P\left(x\right)\left({\hat{v}}^{*}-v^{*}\right)+\nabla V^{*}{\left(x\right)}^{T}G\left(x\right)\left({\hat{u}}^{*}-u^{*}\right). \end{array}$$Then, using Equations (Equation 12) and (Equation 11), it can be derived as
(39)$$\begin{array}{c} \nabla V^{*}{\left(x\right)}^{T}\left(F\left(x\right)+G\left(x\right)u^{*}+P\left(x\right)v^{*}\right)=-x^{T}Qx-U\left(u^{*}\right)+Y^{2}{∥v^{*}∥}^{2}-B_{r}\left(x\right). \end{array}$$Similarly, taking into account Equations (Equation 27) and (Equation 28), the derived results are
(40)$$\nabla V^{*}{\left(x\right)}^{T}G\left(x\right)=2\Psi {\left(tanh^{-1}\left(\left(\psi_{ℑ}-u^{*}\right)/\left(\Psi \right)\right)\right)}^{T},$$
and
(41)$$\nabla V^{*}{\left(x\right)}^{T}P\left(x\right)=2Y^{2}v^{*T}.$$According to Equations (Equation 38)–(Equation 41), Equation (Equation 38) can be rewritten as follows (note: for convenience, $\bar{\omega }-U\left(u^{*}\right)$ and $2Y^{2}v^{*T}{\hat{v}}^{*}-Y^{2}{∥v^{*}∥}^{2}-B_{r}\left(x\right)$ are abbreviated as $\Lambda_{1}$ and $\Lambda_{2}$ below):
(42)$${\dot{L}}_{1}=-x^{T}Qx+\underset{\Lambda_{1}}{\underbrace{\bar{\omega }-U\left(u^{*}\right)}}+\underset{\Lambda_{2}}{\underbrace{2Y^{2}v^{*T}{\hat{v}}^{*}-Y^{2}{∥v^{*}∥}^{2}-B_{r}\left(x\right)}},$$
where
(43)$$\begin{array}{c} \bar{\omega }=2\Psi \left(tanh^{-1}\left(\left(\psi_{ℑ}-u^{*}\right)/\Psi \right)\right)\left({\hat{u}}^{*}-u^{*}\right). \end{array}$$We apply Young’s inequality to Equation (Equation 43). Additionally, considering Equations (Equation 19), (Equation 20), (Equation 27), (Equation 40) and (Equation 41), $\bar{\omega }$ can be formulated as
(44)$$\begin{array}{cc} \bar{\omega } & \leq {∥\Psi (tanh^{-1}\left(\left(\psi_{ℑ}-u^{*}\right)/\Psi \right))∥}^{2}+{∥{\hat{u}}^{*}-u^{*}∥}^{2}\\ \; & =\frac{1}{4}{∥G{\left(x\right)}^{T}\nabla V^{*}\left(x\right)∥}^{2}+{∥{\hat{u}}^{*}-u^{*}∥}^{2}\\ \; & =\frac{1}{4}{∥G{\left(x\right)}^{T}\left(\nabla \delta {\left(x\right)}^{T}W_{c}+\nabla \xi \left(x\right)\right)∥}^{2}\\ \; & \;\;+{∥-\Psi tanh\left(\Gamma \left(x\right)\right)+\Psi tanh\left(\bar{A}\left(x\right)\right)-\xi_{u^{*}}\left(x\right)∥}^{2}. \end{array}$$Furthermore, utilizing Young’s inequality, $\bar{\omega }$ in Equation (Equation 44) further yields
(45)$$\begin{array}{cc} \bar{\omega }\leq & 2{∥-\Psi tanh\left(\Gamma \left(x\right)\right)+\Psi tanh\left(\bar{A}\left(x\right)\right)∥}^{2}+2{∥\xi_{u^{*}}\left(x\right)∥}^{2}\\ & +\frac{1}{2}\left({∥G{\left(x\right)}^{T}\nabla \delta {\left(x\right)}^{T}W_{c}∥}^{2}\right)+\frac{1}{2}\left({∥G{\left(x\right)}^{T}\nabla \xi \left(x\right)∥}^{2}\right)\\ \leq & 4{∥\Psi tanh(\Gamma (x))∥}^{2}+{∥\Psi tanh(\bar{A}\left(x\right))∥}^{2}+2{∥\xi_{u^{*}}\left(x\right)∥}^{2}\\ & +\frac{1}{2}\left({∥G{\left(x\right)}^{T}\nabla \delta {\left(x\right)}^{T}W_{c}∥}^{2}\right)+\frac{1}{2}\left({∥G{\left(x\right)}^{T}\nabla \xi \left(x\right)∥}^{2}\right). \end{array}$$According to Equations (Equation 21) and (Equation 31), the following inequalities can be depicted as
(46)$${∥tanh(\Gamma (x))∥}^{2}=tanh^{2}\left(\Gamma \left(x\right)\right)\leq m$$
*and*
(47)$${∥tanh(\bar{A}\left(x\right))∥}^{2}=tanh^{2}\left(\bar{A}\left(x\right)\right)\leq m.$$Based on Equation (Equation 46) and Assumptions 2 and 3, $\bar{\omega }$ can be expressed as
(48)$$\begin{array}{c} \bar{\omega }\leq 8\Psi^{2}m+\frac{1}{2}G_{M}^{2}\left({ℑ}_{\nabla \delta }^{2}{ℑ}_{W_{c}}^{2}+{ℑ}_{\nabla \xi }^{2}\right)+2b_{\xi_{u^{*}}}^{2}. \end{array}$$By observing Equations (Equation 4) and (Equation 5), it can be concluded that $U(u^{*})>0$. Using Young’s inequality and Equation (Equation 48), the expression of $\Lambda_{1}$ in Equation (Equation 42) can be rewritten as
(49)$$\begin{array}{c} \Lambda_{1}\leq 8\Psi^{2}m+\frac{1}{2}G_{M}^{2}\left({ℑ}_{\nabla \delta }^{2}{ℑ}_{\nabla W_{c}}^{2}+{ℑ}_{\nabla \xi }^{2}\right)+2b_{\xi_{u^{*}}}^{2}. \end{array}$$Similarly, $\Lambda_{2}$ in Equation (Equation 42) can be rewritten as follows (note: from Assumption 1, $B_{r}\left(x\right)$≥0):
(50)$$\begin{array}{cc} \Lambda_{2} & =Y^{2}{∥{\hat{v}}^{*}∥}^{2}-B_{r}\left(x\right)\\ \; & =-Y^{2}{∥v^{*}∥}^{2}+Y^{2}{∥v^{*}∥}^{2}+Y^{2}{∥{\hat{v}}^{*}∥}^{2}-B_{r}\left(x\right)\\ \; & \leq -Y^{2}{∥v^{*}∥}^{2}+Y^{2}{∥v^{*}∥}^{2}+Y^{2}{∥{\hat{v}}^{*}∥}^{2}\\ \; & \leq -Y^{2}{∥v^{*}∥}^{2}+Y^{2}\left({∥v^{*}∥}^{2}+{∥{\hat{v}}^{*}∥}^{2}\right)\\ \; & =\left(1/\left(4Y^{2}\right)\right){∥P{\left(x\right)}^{T}\nabla \delta {\left(x\right)}^{T}\left(W_{c}-{\tilde{W}}_{c}\right)∥}^{2}. \end{array}$$Meanwhile, using Young’s inequality and Assumptions 1 and 3, $\Lambda_{2}$ in Equation (Equation 50) further yields
(51)$$\begin{array}{cc} \Lambda_{2} & \leq \left(1/\left(4Y^{2}\right)\right)P_{M}^{2}{ℑ}_{\nabla \delta }^{2}{∥W_{c}-{\tilde{W}}_{c}∥}^{2}\\ \; & \leq \left(1/\left(2Y^{2}\right)\right)P_{M}^{2}{ℑ}_{\nabla \delta }^{2}\left({ℑ}_{W_{c}}^{2}+{∥{\tilde{W}}_{c}∥}^{2}\right). \end{array}$$Hence, by observing Equations (Equation 49) and (Equation 51), it can be inferred that ${\dot{L}}_{1}$ in Equation (Equation 42) satisfies
(52)$$\begin{array}{cc} {\dot{L}}_{1}\leq & -k_{min}\left(Q\right){∥x∥}^{2}+\left(1/\left(2Y^{2}\right)\right)P_{M}^{2}{ℑ}_{\nabla \delta }^{2}{ℑ}_{W_{c}}^{2}+8\Psi^{2}m+2b_{\xi_{u^{*}}}^{2}\\ \; & +\left(1/2\right)G_{M}^{2}\left({ℑ}_{\nabla \delta }^{2}{ℑ}_{W_{c}}^{2}+{ℑ}_{\nabla \xi }^{2}\right)+\left(1/\left(2Y^{2}\right)\right)P_{M}^{2}{ℑ}_{\nabla \delta }^{2}{∥{\tilde{W}}_{c}∥}^{2}. \end{array}$$Then, the derivative of $L_{2}$ in Equation (Equation 37) along the solution of Equation (Equation 34) is as follows (note: $\alpha {\tilde{W}}_{c}^{T}\left(\zeta /O\right)\xi_{c}$ is abbreviated as $\Lambda_{3}$ below):
(53)$${\dot{L}}_{2}={\tilde{W}}_{c}^{T}{\dot{\tilde{W}}}_{c}=\underset{\Lambda_{3}}{\underbrace{\alpha {\tilde{W}}_{c}^{T}\left(\zeta /O\right)\xi_{c}}}-\alpha {\tilde{W}}_{c}^{T}\zeta \zeta^{T}{\tilde{W}}_{c}.$$Immediately after, using Young’s inequality, $\Lambda_{3}$ can be depicted as
(54)$$\begin{array}{cc} \Lambda_{3} & \leq \frac{\alpha }{2O}\left({∥\zeta^{T}{\tilde{W}}_{c}∥}^{2}+{∥\xi_{c}∥}^{2}\right)\\ \; & \leq \alpha (\frac{1}{2}{\tilde{W}}_{c}^{T}\zeta \zeta^{T}{\tilde{W}}_{c}+\frac{1}{2}{∥\xi_{c}∥}^{2}). \end{array}$$Additionally, with Assumption 3 holding, it can be deduced that ${\dot{L}}_{2}$ in Equation (Equation 53) satisfies
(55)$$\begin{array}{cc} {\dot{L}}_{2} & \leq \frac{1}{2}\left(-\alpha {\tilde{W}}_{c}^{T}\zeta \zeta^{T}{\tilde{W}}_{c}+\alpha {∥\xi_{c}∥}^{2}\right)\\ \; & \leq \frac{1}{2}\left(-\alpha k_{min}\left(\zeta \zeta^{T}\right){∥{\tilde{W}}_{c}∥}^{2}+\alpha {ℑ}_{\xi_{c}}^{2}\right). \end{array}$$Using Equations (Equation 37), (Equation 52) and (Equation 55), $\dot{L}$ can be depicted as
(56)$$\begin{array}{cc} \dot{L}\leq & -k_{min}\left(Q\right){∥x∥}^{2}+\left(1/\left(2Y^{2}\right)P_{M}^{2}{ℑ}_{\nabla \delta }^{2}{ℑ}_{W}^{2}\right)+\left(1/2\right)G_{M}^{2}\left({ℑ}_{\nabla \delta }^{2}{ℑ}_{W}^{2}+{ℑ}_{\nabla \xi }^{2}\right)\\ \; & -\left(1/2\right)\left(\alpha k_{min}\zeta \zeta^{T}-\left(1/Y^{2}\right)P_{M}^{2}{ℑ}_{\nabla \delta }^{2}\right){∥{\tilde{W}}_{c}∥}^{2}+8\Psi^{2}m+2b_{\xi_{u^{*}}}^{2}+\left(\alpha /2\right){ℑ}_{\xi_{c}}^{2}. \end{array}$$Finally, $\dot{L}<0$ is true if x∉℧(x) or ${\tilde{W}}_{c}\notin ℧\left({\tilde{W}}_{c}\right)$, and based on Equation (Equation 36), ℧(x) and $℧({\tilde{W}}_{c})$ can be respectively formulated as
(57)$$\begin{array}{c} ℧\left(x\right)=\left\{∥x∥\leq \sqrt{\frac{\alpha {ℑ}_{\xi_{c}}^{2}+\Xi +A_{1}P_{M}^{2}}{2k_{min}\left(Q\right)}}\right\}, \end{array}$$
and
(58)$$\begin{array}{c} ℧\left({\tilde{W}}_{c}\right)=\left\{∥{\tilde{W}}_{c}∥\leq \sqrt{\frac{\alpha {ℑ}_{\xi_{c}}^{2}+\Xi +A_{1}P_{M}^{2}}{\alpha k_{min}\left(\zeta \zeta^{T}\right)-A_{2}P_{M}^{2}}}\right\}, \end{array}$$
where $\Xi =G_{M}^{2}\left({ℑ}_{\nabla \delta }^{2}{ℑ}_{W_{c}}^{2}+{ℑ}_{\nabla \xi }^{2}\right)+16\Psi^{2}m+4b_{\xi_{u^{*}}}^{2}$, $A_{1}=\left(1/Y^{2}\right){ℑ}_{\nabla \delta }^{2}{ℑ}_{W_{c}}^{2}$ and $A_{2}=\left(1/Y^{2}\right){ℑ}_{\nabla \delta }^{2}$.To summarize, the Lyapunov stability method has been used to demonstrate the state *x* of Equation (Equation 1) and ${\tilde{W}}_{c}$ are UUB, with Equations (Equation 57) and (Equation 58) representing their respective bounds. The proof is complete. □

## 5. Simulation Study

Within this section, two examples are utilized to validate the efficacy of the proposed approach.

### 5.1. Example 1

Consider the F16 aircraft plant used in [28] as
(59)$$\dot{x}=F\left(x\right)+G\left(x\right)u+P\left(x\right)v,$$
where $x\left(t\right)={\left[x_{1},x_{2},x_{3}\right]}^{T}\in R^{3}$ with $x_{0}={\left[1,-1,1\right]}^{T}$ represents the system state vector, where $x_{1},\;x_{2}$ and $x_{3}$ represent the attack angle, the pitch rate, and the elevator deflection angle, respectively. u is control input, v is disturbance input. The internal dynamics, control, and disturbance coefficient matrices are expressed as
$$F\left(x\right)=\left[\begin{array}{c} -1.01887x_{1}+0.90506x_{2}-0.00215x_{3}\\ 0.82225x_{1}-1.07741x_{2}-0.17555x_{3}\\ -x_{3} \end{array}\right],\;G\left(x\right)=\left[\begin{array}{c} 0\\ 0\\ 1 \end{array}\right],\;P\left(x\right)=\left[\begin{array}{c} 0\\ 0\\ -1 \end{array}\right].$$

The control input u is constrained to be greater than −1 and less than 2. Hence, Ψ=1.5 and ℑ=0.5. And then, the danger region is described as a ball with a radius of 0.15 and a center at ${\left[0.3,0.05,-0.05\right]}^{T}$. The y(x) is chosen as
$$\frac{1.5\sqrt{{\left(x_{1}-0.3\right)}^{2}+0.1{\left(x_{2}-0.05\right)}^{2}+1.2{\left(x_{3}+0.05\right)}^{2}}-0.15}{\sqrt{{\left(x_{1}-0.3\right)}^{2}+{\left(x_{2}-0.05\right)}^{2}+25{\left(x_{3}+0.05\right)}^{2}}-0.15}.$$

The z(x) is chosen as
$$\sqrt{{\left(x_{1}-0.3\right)}^{2}+{\left(x_{2}-0.05\right)}^{2}+{\left(x_{3}+0.05\right)}^{2}}-0.15.$$

In addition, substituting Ψ and *ℑ* into Equation (Equation 4), U(u) can be expressed as
(60)$$\begin{array}{cc} U(u) & =2\Psi \left(u-ℑ\right)tanh^{-1}\left(\frac{u-ℑ}{\Psi }\right)+\Psi^{2}\ln_{}(1-\frac{{\left(u-ℑ\right)}^{2}}{\Psi })\\ \; & =3\left(u-0.5\right)tanh^{-1}\left(\frac{u-0.5}{1.5}\right)+2.25\ln_{}(1-\frac{{\left(u-0.5\right)}^{2}}{1.5}). \end{array}$$

Letting $Q=I_{3}$ and Y=2, the cost function for Equation (Equation 62) is formulated as
(61)$$V\left(x\right)=\int_{0}^{\infty }x{\left(t\right)}^{T}Qx\left(t\right)+U\left(u\right)-2^{2}{∥v∥}^{2}+B_{r}\left(x\right)dt,$$
where $B_{r}\left(x\right)=\rho \frac{y(x)}{z(x)}$ represents the CBF and ρ=2.

The activation function is given as $\delta \left(x\right)={\left[x_{1}^{2},x_{1}x_{2},x_{1}x_{3},x_{2}^{2},x_{2}x_{3},x_{3}^{2}\right]}^{T}$ and the CNN weight vector is ${\hat{W}}_{c}={\left[{\hat{W}}_{c1},{\hat{W}}_{c2},{\hat{W}}_{c3},{\hat{W}}_{c4},{\hat{W}}_{c5},{\hat{W}}_{c6}\right]}^{T}$. In addition, the adjustable parameter α is 10, and the original parameters of the CNN are configured as 1. At last, the probing noise $exp\left(-0.1t\right)\left(0.001\right)(sin{\left(t\right)}^{2}cos\left(t\right)+sin{\left(2t\right)}^{2}cos\left(0.1t\right))$ is added to the control input policy for the initial 30 s in order to ensure the persistence of the excitation.

Through simulation experiments, Figure 1, Figure 2, Figure 3, Figure 4, Figure 5, Figure 6 and Figure 7 are obtained. Figure 1 displays that ${\hat{W}}_{c}$ is convergent after the first 10 s, and can know the ideal vector $W_{c}^{*}=[16.4603,-6.5022,$${-4.3910,4.8851,3.7081,11.6158]}^{T}$. Figure 2 displays the convergence of the states $x_{1}$, $x_{2}$, and $x_{3}$. Figure 3 displays the danger region, which is represented by the ball, and the original states are in the danger area. However, the system states controlled by the safe optimal controller bypass this ball, and as the damping coefficient ρ increases, the distance between the system states and the dangerous region becomes larger and larger. Figure 3 shows that as states $x_{1}$, $x_{2}$, and $x_{3}$ gradually approach the danger zone, the convergence of $x_{3}$ is accelerated due to the CBF and cost function. Figure 4 presents the control input u with asymmetric input constraints. The plot reveals that the value of u remains within the specified range, bounded by $u_{max}=2$ and $u_{min}=-1$, providing evidence that the asymmetric input constraints are implemented successfully. Figure 5 presents the disturbance input v. Figure 6 presents the cost function of the system. It can be seen that when the system states confront the danger area, the cost function changes significantly and eventually converges to zero. According to the principle of optimal control, when the cost function converges to zero, the following conclusion can be drawn: The cost function imposes a higher penalty on control actions that do not comply with the asymmetric input constraints and safety constraints. Therefore, when the cost function converges to zero, the system finds the optimal control actions that satisfy all the constraints.

In order to further show the efficiency of the presented method, Equation (Equation 4) is redefined as $u^{T}Ru$ (where *R* = $I_{1}$), and the simulation results are illustrated in Figure 7. Subsequently, Figure 4 illustrates the control input, which is restricted to the limits of −1 to 2. This can be observed by comparing it with Figure 7, where the input is clearly outside this range.

### 5.2. Example 2

We consider the nonlinear system as
(62)$$\dot{x}=F\left(x\right)+G\left(x\right)u+P\left(x\right)v,$$
where $x\left(t\right)={\left[x_{1},x_{2}\right]}^{T}\in R^{2}$ with $x_{0}={\left[1,-1\right]}^{T}$ represents the system state vector; the internal dynamics, control, and disturbance coefficient matrices are expressed as
$$F\left(x\right)=\left[\begin{array}{c} -\frac{1}{2}x_{1}+x_{2}\\ -2x_{2}cos\left(2x_{1}\right) \end{array}\right],\;G\left(x\right)=\left[\begin{array}{c} 0\\ -x_{1} \end{array}\right],\;P\left(x\right)=\left[\begin{array}{c} 0\\ x_{1} \end{array}\right].$$

Just like F16, the control input u is subject to an asymmetrical boundary, with a lower bound of −1 and an upper bound of 3, establishing its limits. Hence, Ψ=2 and ℑ=1. And then, the danger region is described as a circle with radius =0.1, and the center of the circle is ${\left[0.19,-0.12\right]}^{T}$. The y(x) is chosen as
$$atan(\frac{1}{\sqrt{{\left(x_{1}-0.19\right)}^{2}+{\left(x_{2}+0.12\right)}^{2}}-0.1}).$$

The z(x) is chosen as
$$\sqrt{{\left(x_{1}-0.19\right)}^{2}+{\left(x_{2}+0.12\right)}^{2}}-0.1.$$

In addition, substituting Ψ and *ℑ* into Equation (Equation 4), U(u) can be expressed as
(63)$$\begin{array}{cc} U(u) & =2\Psi \left(u-ℑ\right)tanh^{-1}\left(\frac{u-ℑ}{\Psi }\right)+\Psi^{2}\ln_{}(1-\frac{{\left(u-ℑ\right)}^{2}}{\Psi })\\ \; & =4\left(u-1\right)tanh^{-1}\left(\frac{u-1}{2}\right)+4\ln_{}(1-\frac{{\left(u-1\right)}^{2}}{2}). \end{array}$$

Letting $Q=I_{2}$ and Y=1.35, the cost function for Equation (Equation 62) is formulated as
(64)$$V\left(x\right)=\int_{0}^{\infty }x{\left(t\right)}^{T}Qx\left(t\right)+U\left(u\right)-1.35^{2}{∥v∥}^{2}+B_{r}\left(x\right)dt,$$
where $B_{r}\left(x\right)=\rho \frac{y(x)}{z(x)}$ represents the CBF and ρ=0.3.

Then, the CNN presented as Equation (Equation 18) is applied to address the HJI equation for Equation (Equation 62). The activation function is given as $\delta \left(x\right)={\left[x_{1}^{2},x_{1}x_{2},x_{2}^{2},x_{1}^{4},x_{1}^{3}x_{2},x_{1}^{2}x_{2}^{2},x_{1}x_{2}^{3},x_{2}^{4}\right]}^{T}$ and the CNN weight vector is ${\hat{W}}_{c}={\left[{\hat{W}}_{c1},{\hat{W}}_{c2},{\hat{W}}_{c3},{\hat{W}}_{c4},{\hat{W}}_{c5},{\hat{W}}_{c6},{\hat{W}}_{c7},{\hat{W}}_{c8}\right]}^{T}$. In addition, the adjustable parameter α is 20, the original parameters of the CNN are configured as 1. At last, the probing noise $exp\left(-0.001t\right)(-0.1\left(sin{\left(t\right)}^{2}cos\left(t\right)+sin{\left(t\right)}^{5}+sin{\left(2t\right)}^{2}cos\left(0.1t\right)+sin{\left(-1.2t\right)}^{2}cos\left(0.5t\right)\right)$ is added to the control input policy for the initial 30 s.

Through simulation experiments, Figure 8, Figure 9, Figure 10, Figure 11, Figure 12, Figure 13 and Figure 14 are obtained. Figure 8 displays that ${\hat{W}}_{c}$ is convergent after the first 10 s, and can know the ideal vector $W_{c}^{*}=[84.6487,-12.2017,$${9.5269,11.7425,-3.0924,3.4273,-0.5533,2.0591]}^{T}$. Figure 9 displays the convergence of the states $x_{1}$ and $x_{2}$. Figure 10 illustrates the relationship between the system states and the dangerous area, revealing that increasing the damping factor ρ leads to a greater distance between the system states and the dangerous zone. Evidently, system states $x_{1}$ and $x_{2}$ with a safe and optimal controller take an alternate route to avoid the dangerous region, while the conventional optimal controller cannot circumvent the dangerous region. As can be seen from Figure 10, when states $x_{1}$ and $x_{2}$ gradually approach the danger zone, the convergence speed of $x_{2}$ is accelerated due to the influence of CBF and cost function and obtains an optimal trajectory around the danger zone again. Figure 11 shows input u with asymmetric input constraints. The plot reveals that the value of u remains within the specified range, bounded by $u_{max}=3$ and $u_{min}=-1$, providing evidence that the asymmetric input constraints are implemented successfully. Figure 12 presents disturbance input v. Figure 13 presents the cost function of the system. It can be seen that the cost function eventually converges to zero. Similar to the linear system, when the cost function converges to zero, it can be concluded that the system finds the optimal control action that satisfies the asymmetric input constraints and safety constraints.

In this paper, asymmetric input constraints and unmatched disturbances are applied to nonlinear safety-critical systems for the first time, and Equation (Equation 4) is used to handle the asymmetric input constraints. To further demonstrate the efficacy of the presented algorithm, as in articles [14,16,28], (Equation 4) is redefined as $u^{T}Ru$ (where *R* = $I_{1}$) and the simulation results are shown in Figure 14. Subsequently, the control input in Figure 11 is constrained to fall within the limits of −1 to 3, as can be observed by comparing it with Figure 14, while the input in Figure 14 is clearly outside this range.

## 6. Conclusions

The safe and optimal control problem of the nonlinear CT safety-critical systems with asymmetric input constraints and unmatched disturbances was addressed. Firstly, the new non-quadratic form function was considered for addressing the issue of asymmetric input constraints. Then, the control design was transformed into the two-player ZSG problem to handle unmatched disturbances. In order to obtain the optimal controller for safety, the combination of the CBF and cost function was directly used to penalize unsafe behavior. Moreover, the CNN was applied to reduce the computational complexity of dual actor–critic network. The effectiveness of the proposed method was validated by the simulation results.

## Figures and Tables

**Figure 1 entropy-25-01101-f001:**
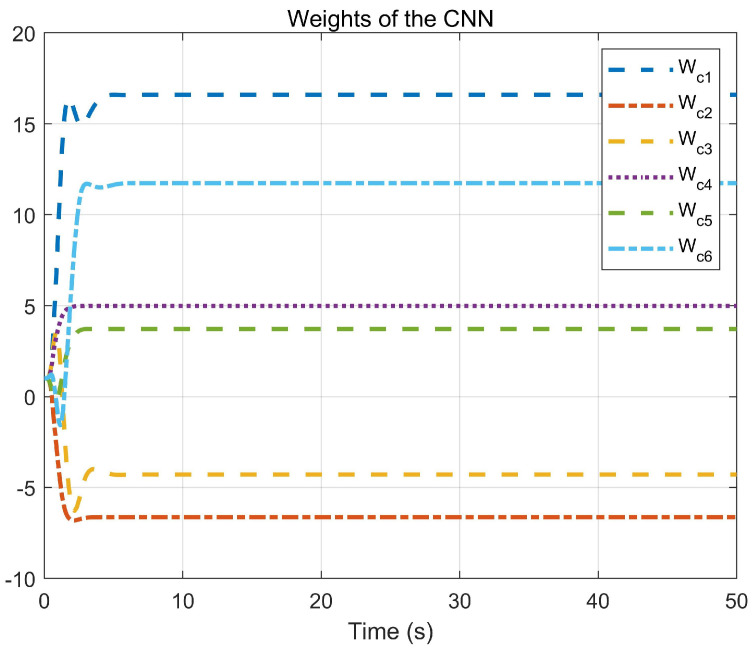
Convergence of the CNN weights.

**Figure 2 entropy-25-01101-f002:**
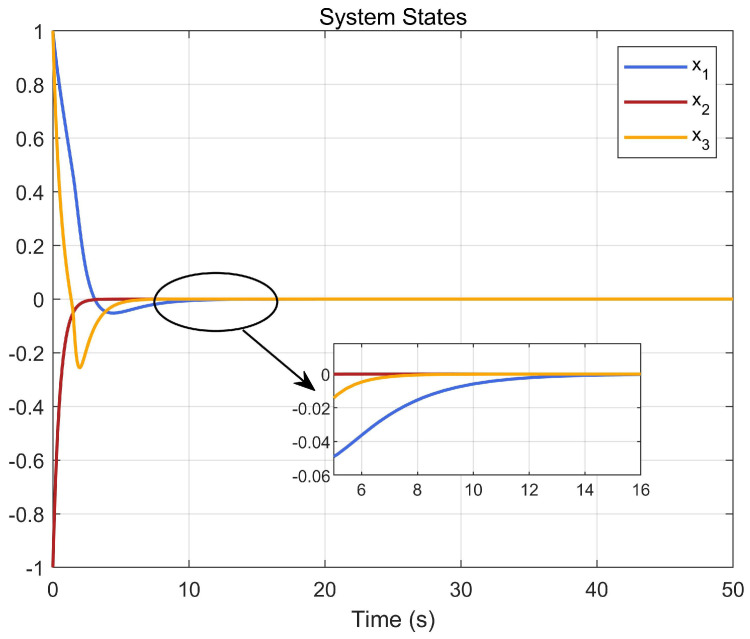
Convergence of system states $x_{1}$, $x_{2}$, and $x_{3}$.

**Figure 3 entropy-25-01101-f003:**
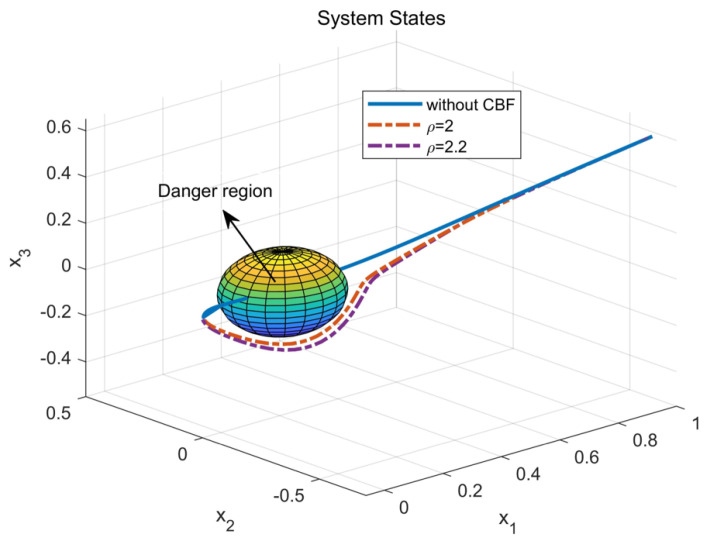
The comparison between the safe and unsafe states.

**Figure 4 entropy-25-01101-f004:**
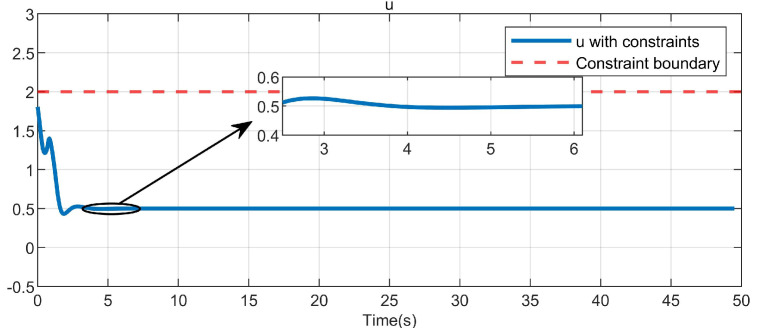
Control input in the system.

**Figure 5 entropy-25-01101-f005:**
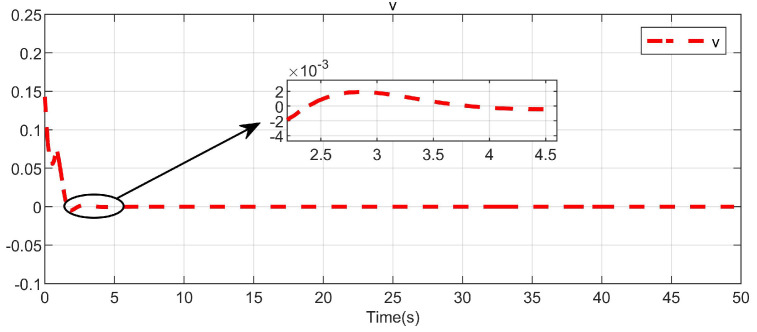
Disturbance input in the system.

**Figure 6 entropy-25-01101-f006:**
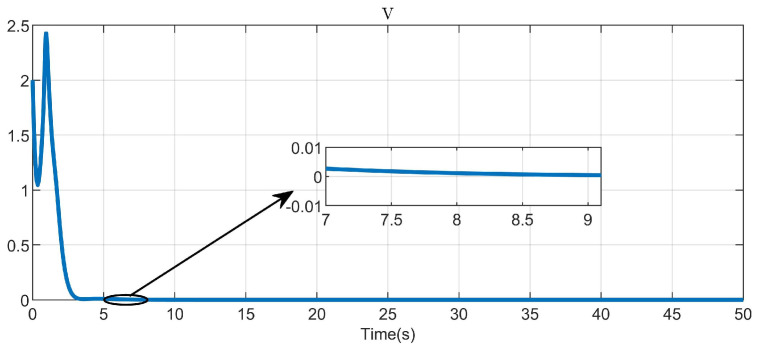
The cost function of the system.

**Figure 7 entropy-25-01101-f007:**
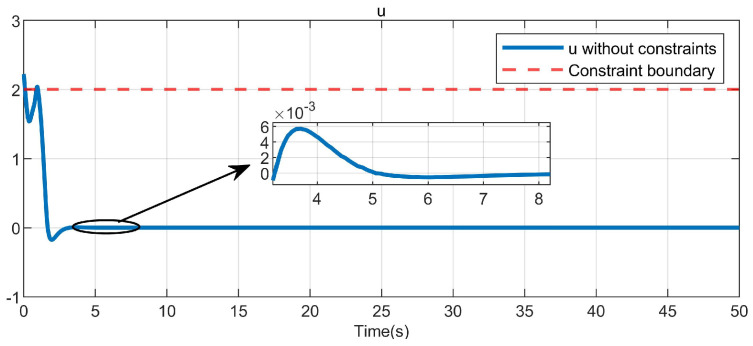
Control input without asymmetric input constraints.

**Figure 8 entropy-25-01101-f008:**
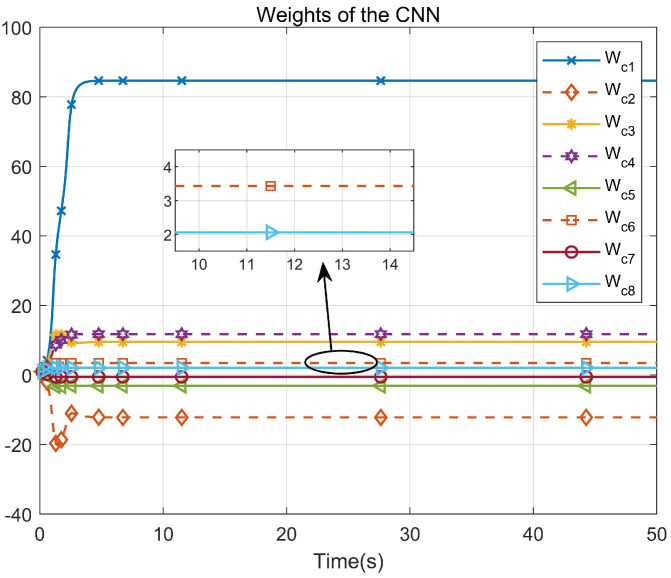
Convergence of the CNN weights.

**Figure 9 entropy-25-01101-f009:**
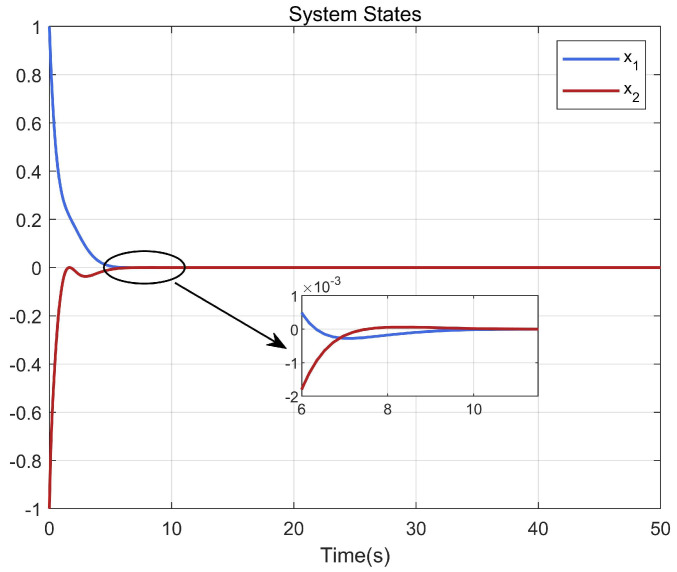
Convergence of system states $x_{1}$ and $x_{2}$.

**Figure 10 entropy-25-01101-f010:**
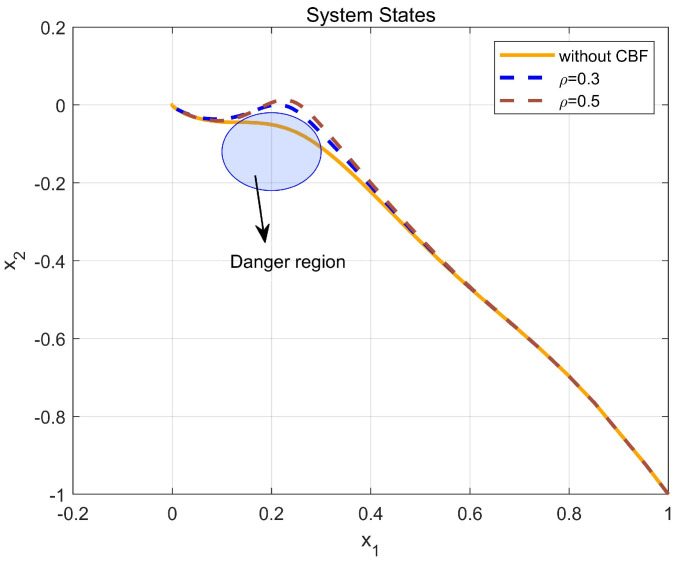
The comparison between the safe and unsafe states.

**Figure 11 entropy-25-01101-f011:**
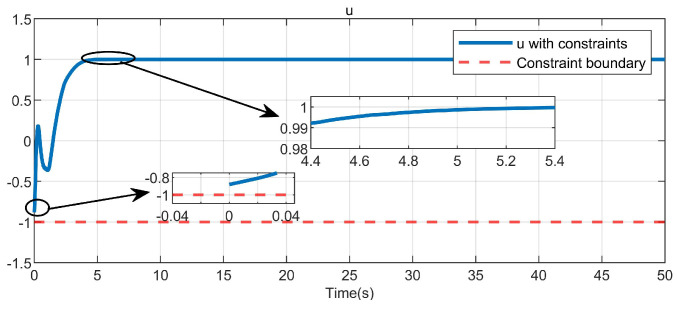
Control input in the system.

**Figure 12 entropy-25-01101-f012:**
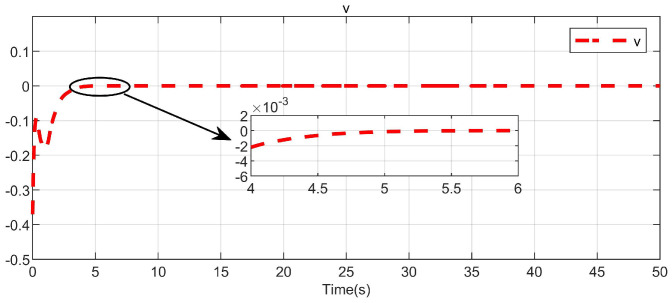
Disturbance input in the system.

**Figure 13 entropy-25-01101-f013:**
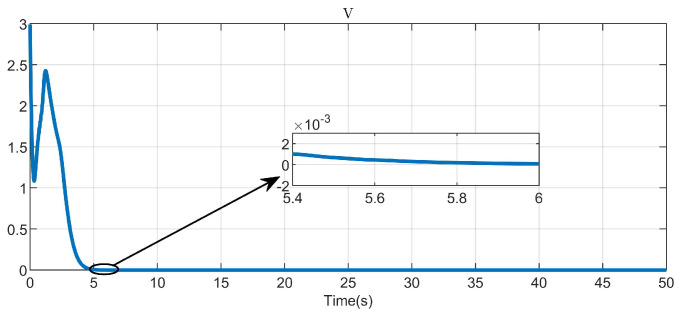
The cost function of the system.

**Figure 14 entropy-25-01101-f014:**
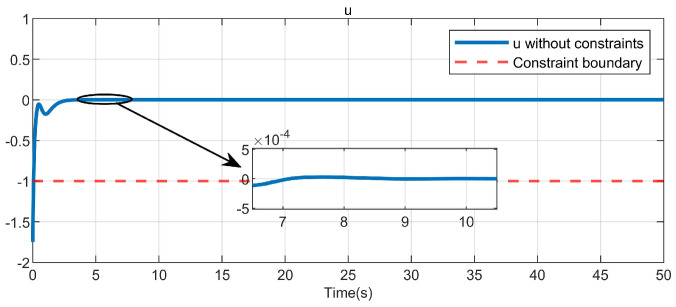
Control input without asymmetric input constraints.

## Data Availability

The authors can confirm that all relevant data are included in the article.

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
