# Peer review of "Critic Learning-Based Safe Optimal Control for Nonlinear Systems with Asymmetric Input Constraints and Unmatched Disturbances"

_entropy, 2023, doi:10.3390/e25071101_

Round 1

Reviewer 1 Report

1. What is the major difference between the authors' previous published paper "Robust Trajectory Tracking Control for Continuous-Time Nonlinear Systems with State Constraints and Uncertain Disturbances" and this paper? 

2. Experiment is missing, and it is strongly recommended to add the experiment to verify the results in Section 5. 

3. All the equations should be simplified and explained in details in the paper, otherwise it is very difficult for readers to follow the equations. 

4. In section 5, the authos gave two examples, but it is recommended to compare these two examples and show what the relations between these two examples are. 

Minor editing of English language required

Reviewer 2 Report

This paper deals with the safe and optimal control problem of the nonlinear continuous-time safety-critical systems with asymmetric input constraints and unmatched disturbances.

In the first part, a non-quadratic form function is considered for addressing the issue of asymmetric input constraints. Then, the control design is remade into a two-player zero-sum game problem to handle unmatched disturbances.

Finally, to obtain the optimal controller for safety, the control barrier function and cost function are directly integrated to penalize unsafe behavior. Furthermore, a single critic neural network is used to reduce the computational complexity.

The paper is well written and is very clear in its structure and exposition.

I would suggest integrating two references in the introduction part in order to improve the work.

1)     When it comes to the application of drones, the problem of safety in the presence of unmodeled dynamics or disturbances has recently been addressed.

Bianchi, D.; Di Gennaro, S.; Di Ferdinando, M.; Acosta Lùa, C. Robust Control of UAV with Disturbances and Uncertainty Estimation. Machines 2023, 11, 352. https://doi.org/10.3390/machines11030352

2) The use of neural networks integrated with Lyapunov's theory was preliminarily treated with application in the automotive sector for critical situations, in the presented paper it is dealt with in an even more organic way and therefore it is a way to enhance the work.

Bianchi, D.; Borri, A.; di Benedetto, M.D.; di Gennaro, S. Active Attitude Control of Ground Vehicles with Partially Unknown Model. IFAC-PapersOnLine 2020, 53, 14420–14425.

The simulations and figures are very clear, I would add some considerations, graphs, tables on the functional to better quantify the performance.

Round 2

Reviewer 1 Report

In my previous comment 2 (The experiment is missing, and it is strongly recommended to add the experiment to verify the results in Section 5), I was not talking about the simulation experiment, I was talking about the real experiment. The authors need to add the experiment in the paper to verify the results provided in Section 5. 

Minor editing of English language required

Reviewer 2 Report

The paper has been improved and it is ready for publication.

some parts could probably be improved in the expressive form.